# Morpho-Quantitative Traits and Interrelationships between Environmental Factors and *Phytophthora infestans* (Mont.) de Bary Attack in Tomato

Roxana Alexandra Sabo [1], Csaba-Pál Racz [2], Ioan Oroian [1], Petru Burduhos [1,*], Camelia Manuela Mîrza [3], Claudia Balint [1], Cristian Mălinaș [1] and Antonia Cristina Maria Odagiu [1]

[1] Faculty of Agriculture, University of Agricultural Sciences and Veterinary Medicine Cluj-Napoca, 3–5 Calea Mănăștur, 400372 Cluj-Napoca, Romania; ioan.oroian@usamvcluj.ro (I.O.); vbalintc@gmail.com (C.B.); malinas.cristian@gmail.com (C.M.); aodagiu@gmail.com (A.C.M.O.)

[2] Faculty of Chemistry and Chemical Engineering, Babeș Bolyai University, 11 Arany Janos Street, 400028 Cluj-Napoca, Romania; csaba.racz@ubbcluj.ro

[3] Faculty of Medicine, "Iuliu Hațieganu" University of Medicine and Pharmacy, 8 Victor Babeș Street, 400012 Cluj-Napoca, Romania; camelia.mirza@umfcluj.ro

[*] Correspondence: petru.burduhos@usamvcluj.ro or petru.burduhos@gmail.com; Tel.: +40-745297993

**Abstract:** The growing imperative for sustainable development generates research in the field and explores innovative approaches to manage vegetable crops to reduce the usage of synthetic inputs commonly associated with conventional practices as well as to promote the undertaking of organic solutions. Tomatoes are universally recognized as a highly significant and popular fruit vegetable due to their large use palette. Trials were carried out in the Gâlgău area, Transylvania, Romania. Bifactorial experiments were organized to quantify the influence of fertilization and treatments on the morpho-productive and quantitative traits of the Ruxandra tomato cultivar and on *Phytophthora infestans* levels of infection. The use of mixed fertilization resulted in the best performances of morphological traits (highest leaf area, highest number of leaves and fruits, highest chlorophyll content) and part of the productive traits (highest content of dry matter), and the lowest levels of infection in cv. Ruxandra, while the use of NPK soil fertilization led to the best performances in fruit weight and NUE, whatever the administered treatment. The treatment with herbal extracts showed a similar efficacy in increasing the averages of morpho-productive traits and diminishing the *P. infestans* level of infection compared with the conventional treatment. The treatment efficacy was mainly influenced by temperature and relative air humidity, regardless of the fertilization strategies applied.

**Keywords:** sustainable food; *Phytophthora infestans*; herbal extracts; tomato crop; sustainable development

## 1. Introduction

In the context of increasing concerns for promoting sustainable development, innovative solutions for vegetable crop management are investigated in order to mitigate the synthesis inputs used in conventional practices and increase the extent of practicing organic solutions [1,2]. A series of factors result in varying levels of significance among cultivated crops in human nutrition. Among the crops covering the largest cultivation area worldwide, tomatoes (*Solanum lycopersicum* L.) occupy the 18th place [1–3]. In this respect, it is notable that tomatoes are globally recognized as a highly significant and popular fruit vegetable, because of their use both for fresh consumption and raw material in the food industry [4]. In 2020, global tomato production was almost 190 metric tons [5]. The study of plant morpho-productive traits (leaf area, number of leaves, number of fruits, chlorophyll content dry matter, fruit weight, nitrogen use efficiency) in environmental-specific conditions is of interest because it may contribute to developing strategies concerning the improvement of plant yield and quality [6].

Additionally, tomatoes are susceptible to the attack of several pathogens, but not all diseases carry the same impact [7]. Among tomato pathogens, *Phytophthora infestans* (Mont.) de Barry is still considered as one of the most harmful due to its high capacity of spreading and the fact that climatic conditions such as temperature, rainfall regimen, and air humidity favor the developmental phases of the fungus [8]. Plant and pathogen interactions are characterized by complex mechanisms and environmental conditions are considered as influential to both the condition and defense mechanisms of the plant [9–13]. Environmental factors also influence the pathogen's survival, considering their role in oospore germination [13,14]. According to research performed by Meno et al. (2016) on the effect of climatic conditions influencing the *P. infestans* sporangia levels in the air, their increase is correlated with temperature and relative humidity [15]. Favorable conditions for oospore germination were identified with a constant relative humidity over 90% and temperature frames in the range of 10–15 °C, while the infection of plant tissue occurs within the range of 20–25 °C [11,14–19].

Great concern regarding the excessive use of agrochemicals in phytosanitary treatments administered to vegetables in general, and to tomato cultures in particular, against severe diseases such as late blight, drives preoccupations and research in the field towards ecological solutions, which may provide satisfactory outputs [20–24]. In this respect, studies show that essential oils originating from aromatic plants have satisfactory antibacterial and antifungal action, mainly due to their composition. Among these plants, of interest may be lavender, thyme, and rosemary. The majority of studies identified linalool (20–45%) and linalyl acetate (25 to 46%) as the main components of lavender essential oil. Research concerning lavender properties mainly emphasizes their antifungal, antibacterial, and antioxidant activities [20–27]. Sarkhosh et al. (2018) identified in *Lavandula angustifolia* L. about 19 detectable compounds. According to an in vitro trial, they observed a linear relationship between *Phytophthora palmivora* mycelial growth and the application rate of lavender essential oil. They found that the administration of *L. angustifolia* essential oil at high concentrations completely inhibited *P. palmivora* mycelial growth [21]. According to Puškárová et al. (2017), *L. angustifolia* exhibits fungistatic activity, only in the vapor phase, on different fungal strains (*Cladosporium cladosporoides*, *Aspergillus fumigatus*, *Chaetomium globosum*, *Penicillium chrysogenum*) [23]. Soylu et al. (2006) reported total inhibition when in vitro fumigation tests against *P. infestans* mycelium growth were performed with *L. officinalis* essential oil [28] In rosemary (*Rosmarinus officinalis* L.), Hussain et al. (2010) found 20 components with a predominance of 1,8-cineol and α-pinene [29]. Ben Kaab et al. (2019) emphasize the plant's antifungal and herbicidal action. An evaluation of the antifungal activity of rosemary essential oils extracts in a bioassay involving *Fusarium oxysporum*, *Fusarium culmorum*, and *Penicillium italicum* found that *F. oxysporum* and *F. culmorum* are more sensitive than *P. italicum* [30]. When *R. officinalis* essential oil was used against *P. infestans* infection, a late blight severity reduction by 90% was reported in a greenhouse environment [31]. In *Satureja hortensis* L. essential oil, thymol (0.3–28.2%), γ-terpinene (15.30–39%), and carvacrol (11–67%) were identified as majority compounds [31–35]. Testing the antifungal activity of *S. hortensis* essential oil, Güllüce et al. (2003) found that it has a great potential against 12 fungi isolates [33].

Considering the importance of tomato production in the food industry in the context of a sustainable approach, our research was carried out with the aim of identifying the influence of the agricultural inputs of fertilization (foliar with a mineral complex, and soil with complex N15:P15:K15) and phytosanitary treatments (with azoxistrobin, lavender, thyme, and rosemary herbal extracts) on plant morphological traits, and the interrelationships between environmental factors and *P. infestans* infection in tomato. The morphological traits of plants (number of leaves, surfaces, etc.) represent an important aspect in crop management because they substantially contribute to an adequate photosynthesis process with an essential role in ensuring increased productivity [36,37]. Moreover, the environmental factors are known for their influence on *P. infestans* infection potential.

## 2. Materials and Methods

### 2.1. Location

The experiments took place in the years 2021 and 2022, respectively, in the commune of Gâlgău (47°17′03″ N, 23°40′34″ E) with the pedoclimatic characteristics of Transylvania, Romania, on a 400 m$^2$ area, taking into account the traits of the location. The annual mean temperature in the area frames within 7.5–8 °C, the precipitations within 700–900 mm, while wind predominant direction is West [38]. The soil belongs to Luvisol group and Luvosol subgroup [39]. The climatic indicators were monitored during the experimental period April–June corresponding to the two years, as well as the experimental conditions (administration of fertilizers and treatments against *P. infestans*). The data related to the morpho-productive characteristics of the tomato crop were collected, as well as the levels of infection produced by *P. infestans* depending on the level of agricultural inputs administered (fertilizers and phytosanitary treatments). The biological material studied is represented by the tomato (*S. lycopersicum*) Ruxandra cultivar (Agrosel, Romania).

### 2.2. Experimental Design

The studies on the influence of fertilization and phytosanitary treatments on the morpho-productive and quantitative properties of cv. Ruxandra, and on the effectiveness of conventional and treatments with herbal extracts in the fight against *P. infestans*, were organized as bifactorial experiments. Factor A—fertilization, common for both experiments, comprises 4 treatments ($a_1$: unfertilized, $a_2$: soil fertilization, $a_3$: foliar fertilization and $a_4$: mixed fertilization, foliar and soil fertilization). For the experiment concerning morpho-productive and quantitative properties, Factor B—phytosanitary treatment, comprises 3 treatments ($b_1$: untreated control, $b_2$: conventional treatment, $b_3$: herbal treatment with mixture of commercially available aqueous extracts 5%, thyme 1% and rosemary 5%, 40:30:30, *v/v/v*). The combination of the two factors resulted in 12 experimental variants, with 20 plants/variant/two years experimental period in 3 replicates R (R1—6 plants, R2—6 plants, R3—8 plants), organized in randomized blocks. For the experiment concerning the effectiveness of treatments against pathogen, Factor B—phytosanitary treatment, comprises 6 treatments: b1: untreated control, b2: conventional treatment, b3: herbal treatment with aqueous lavender extract 5%, $b_4$: herbal treatment with aqueous extract of thyme 1%, $b_5$: herbal treatment with aqueous extract of rosemary 5%, $b_6$: herbal treatment with mixture of aqueous extracts of lavender 5%, thyme 1% and rosemary 5%, 40:30:30, *v/v/v*). The combination of both factors resulted in 24 experimental variants organized in randomized blocks, with 20 plants/variant/two years experimental period in 3 replicates R (R1—6 plants, R2—6 plants, R3—8 plants), and organized in randomized blocks.

In both experimental years, the planting of the seedlings in the field was performed in mid-April, at a density of 2 plants/m$^2$. Soil fertilization was performed with complex fertilizer N15:P15:K15 (Mifalchim, Onești, Romania) administered in dose of 40 g/m$^2$. The first fertilization was applied two weeks after plantation, and 3 times after, at a 30-day interval. For foliar fertilization, YaraVita mineral complex (Yara International; Oslo, Norway) was administered at a dose of 2.5 L/ha, before flowering, twice, in the beginning and end of May. Mixed fertilization was performed foliarly with YaraVita, and in the soil, using N15:P15:K15. Both fertilizers were administered at the same doses, with the same frequency and in the same periods as described for simple soil and foliar fertilization.

The conventional treatment against *P. infestans* was performed using Allstar 250 SC (JT Agro Ltd., Maidenhead, UK) with azoxystrobin as active compound (250 mg/L). Lavender oil 100% (Fares, Romania), thyme oil 100% (Fares, Orăștie, Romania), and rosemary oil 100% (Fares, Romania) were used for treatments with herbal extracts. The 5% (*v/v*) aqueous solutions were prepared for the herbal treatment. Three treatments with conventional and herbal products were administered, first in mid-May, and last in mid-June. First watering was administered one week after plantation. The frequency of watering was once at 10 days (5 L/m$^2$) in the first part of crop development, and twice in the last month, at the same doses.

The morphometric determinations refer to leaf surface, the number of leaves and the number of fruits, while the quantitative ones refer to fruit weight, the efficiency of nitrogen use, fruit dry matter and leaf chlorophyll content. Leaf surface ($cm^2$) was determined by the leaf parameters method [40]. Nitrogen use efficiency was calculated as function of nitrogen content in fertilized and non-fertilized plants, and nitrogen administration rate [41]. The laboratory analyses consist of determinations of dry matter/water content by the gravimetric method and chlorophyll by the non-destructive method with SPAD-502 [42,43].

The methodology used for the determination of the levels of infection provides the attack degree calculation. It involves twice-a-week observations of infection intensity and frequency [44]. Infection intensity and frequency of *P. infestans* were recorded, in order to calculate the attack degree (AD), according to the formula [44,45]:

$$AD = F \times I/100 \tag{1}$$

where

AD—attack degree, %.
F—attack frequency; it is the relative value of the number of plants infected by *P. infestans* related to the number of plants observed. The value of frequency is obtained by direct observations on a number of plants (10 by experimental variant in our case).
I—attack intensity; it is the value by which the degree of coverage or extension of the infection is given, reporting the infected surface compared to the total observed surface.

The means of the climatic parameters, the morphometric traits, production traits of plants and the level of infection were calculated for each experimental year. Because no significant differences ($p > 0.05\%$) in means were recorded, results are expressed as averages over the entire experimental period.

### 2.3. Statistics

The STATISTICA v.8 for Windows and "XLSTAT" (https://www.xlstat.com/en/, accessed on 28 October 2023) programs were used to apply statistical tools. Descriptive statistics were implemented to calculate the averages and dispersion parameters (standard deviation and variability) of all analyzed indicators, respectively: morphological parameters (leaf surface, number of leaves, number of fruits), productive characteristics (fruit weight, nitrogen utilization efficiency), quantitative parameters (leaf chlorophyll content, fruit dry matter) and pathogen levels of infection. ANOVA on the basis of the *t*-test was used for the calculation of the significance of differences in the means of plants traits and levels of infections, function of treatments and fertilization. To emphasize the common effects of fertilization and treatments on the levels of *P. infest teans* infection in cv. Ruxandra, the Least Square Differences Test ($LSD_{5\%}$) was applied. In order to identify the interrelationships between agricultural inputs represented by fertilizers and phytosanitary treatments, levels of infection, and environmental factors (temperature, precipitation, relative air humidity, wind speed), exploratory analysis (factor analysis through its component PCA Principal Components Analysis) was used. In order to test the possibility of applying PCA, the Keiser–Meyer–Olkin (KMO) and Bartlett tests were implemented in all cases, for which threshold values above 0.500 and $p < 0.01$ are considered, respectively [46]. Multivariate analysis was used to highlight the interrelationships between the morpho-productive parameters and the quantitative production features, by calculating the multiple correlations between the mentioned indicators in the context of specific agricultural inputs represented by fertilizers and phytosanitary treatments applied.

### 3. Results and Discussions

#### 3.1. Environmental Factors in Studied Area

Thus, the temperature has an average of 17.1 °C, with a minimum of 3.90 °C and a maximum of 26.78 °C. The precipitation regime is characterized by an average equal to

3.50 mm, which corresponds to the sum of 181.80 mm, with a minimum equal to 0.5 mm and a maximum equal to 36 mm; the relative air humidity shows the average of 71.01% with a minimum equal to 51.10% and a maximum equal to 92.40%, and the wind speed an average equal to 9.07 km/h with a minimum equal to 3, 70 km/k and a maximum equal to 20.28 km/h (Table 1).

**Table 1.** The basic statistics for climatic parameters reported in the experimental field, April–July, by 2021–2022.

| Issue | N | X | S | Min. | Max. | s |
|---|---|---|---|---|---|---|
| Temperature (°C) | 122 | 17.15 | 2092.17 | 3.90 | 26.78 | 5.42 |
| Rainfall regimen, mm | 52 | 3.50 | 181.80 | 0.50 | 36.00 | 5.96 |
| Relative humidity, % | 122 | 71.01 | 8662.80 | 51.10 | 92.40 | 8.95 |
| Wind velocity, % | 122 | 9.07 | 1107.07 | 3.70 | 20.28 | 3.26 |

N—number of days; X–mean; S—sum; s—standard deviation.

### 3.2. The Impact of Conventional and Non-Conventional Agricultural Inputs on the Morpho-Productive Properties of Tomatoes

As shown, the average foliar area was the lowest in unfertilized and untreated plants; no statistical difference in average foliar area was observed between NPK-soil-fertilized and foliar-fertilized untreated plants and untreated plants fertilized with mixed fertilizers showed the highest foliar area (Table 2).

### 3.2.1. Untreated Plants

The average numbers of leaves and fruits in unfertilized plants do not significantly differ from those reported when NPK soil fertilization is applied. Additionally, there is no significant difference between the average number of foliar-fertilized leaves and those obtained when no fertilization is performed and NPK is applied, on the one hand, and when mixed fertilization was administered, on the other hand. When mixed fertilization was administered, the average leaf and fruit numbers differ from those corresponding to the lack of fertilization, or NPK fertilization, but not from those foliar fertilized. Differences were found in the average leaf number of mixed-fertilized plants and those not fertilized, and NPK fertilized. No statistical differences in chlorophyll contents are noticeable in unfertilized, NPK-soil-fertilized, and foliar-fertilized plants. There are differences in the chlorophyll content of mixed-fertilized plants on the one hand, and unfertilized, NPK-soil-fertilized, and foliar-fertilized plants.

### 3.2.2. Conventional Phytosanitary Treatment

The lowest averages of all analyzed traits correspond to the unfertilized variant. No statistical difference in average foliar area is reported in NPK-soil-fertilized and foliar-fertilized plants. However, differences are reported in the average foliar area of unfertilized plants on the one hand, and NPK-soil-fertilized, foliar-fertilized, and mixed-fertilized plants. Regarding the average leaf number, average fruits number and average chlorophyll content, no statistical differences were observed between unfertilized, NPK-soil-fertilized and foliar-fertilized plants. In the abovementioned trait differences in the means, the only ones observed are between mixed-fertilized plants, on the one hand, and those corresponding to other fertilization variants, on the other hand.

**Table 2.** The evolution of some morpho-nutritional traits in cv. Ruxandra, and function of agricultural inputs.

| Trait | Phytosanitary Treatment | Fertilization | N | X | s | CV (%) |
|---|---|---|---|---|---|---|
| Foliar area, cm$^2$ | | Unfertilized | 20 | 52.80 a | 7.53 | 14.26 |
| | | NPK soil fertilization | 20 | 56.10 b | 6.70 | 11.95 |
| | | Foliar fertilization | 20 | 57.55 b | 5.79 | 10.06 |
| | | Mixed fertilization | 20 | 60.40 c | 5.97 | 9.88 |
| Leaf number | | Unfertilized | 20 | 20.45 a | 2.68 | 13.13 |
| | | NPK soil fertilization | 20 | 21.30 a | 2.87 | 13.46 |
| | Untreated | Foliar fertilization | 20 | 22.30 ab | 2.25 | 10.09 |
| | | Mixed fertilization | 20 | 23.25 b | 3.55 | 15.28 |
| Fruit number | | Unfertilized | 20 | 5.65 a | 1.42 | 25.21 |
| | | NPK soil fertilization | 20 | 5.60 a | 0.94 | 16.79 |
| | | Foliar fertilization | 20 | 6.45 b | 1.57 | 24.37 |
| | | Mixed fertilization | 20 | 6.75 b | 1.62 | 23.97 |
| Chlorophyll, SPAD | | Unfertilized | 20 | 54.70 a | 5.19 | 9.49 |
| | | NPK soil fertilization | 20 | 55.30 a | 4.50 | 8.13 |
| | | Foliar fertilization | 20 | 55.35 a | 3.90 | 7.04 |
| | | Mixed fertilization | 20 | 57.70 b | 2.52 | 4.36 |
| Foliar area, cm$^2$ | | Unfertilized | 20 | 53.90 a | 6.92 | 12.84 |
| | | NPK soil fertilization | 20 | 58.10 b | 5.96 | 10.26 |
| | | Foliar fertilization | 20 | 58.30 b | 5.41 | 9.28 |
| | | Mixed fertilization | 20 | 62.55 c | 5.31 | 8.48 |
| Leaf number | | Unfertilized | 20 | 21.45 a | 3.22 | 15.01 |
| | | NPK soil fertilization | 20 | 22.75 a | 2.36 | 10.37 |
| | Conventional | Foliar fertilization | 20 | 23.00 a | 2.70 | 11.72 |
| | | Mixed fertilization | 20 | 23.95 ab | 3.14 | 13.10 |
| Fruit number | | Unfertilized | 20 | 6.15 a | 1.53 | 24.90 |
| | | NPK soil fertilization | 20 | 6.55 a | 1.64 | 25.00 |
| | | Foliar fertilization | 20 | 6.90 a | 1.41 | 20.44 |
| | | Mixed fertilization | 20 | 7.25 b | 1.71 | 23.63 |
| Chlorophyll, SPAD | | Unfertilized | 20 | 56.45 a | 4.66 | 8.26 |
| | | NPK soil fertilization | 20 | 55.85 a | 4.30 | 7.69 |
| | | Foliar fertilization | 20 | 56.30 a | 4.23 | 7.52 |
| | | Mixed fertilization | 20 | 58.50 b | 3.03 | 5.19 |
| Foliar area, cm$^2$ | | Unfertilized | 20 | 52.50 a | 6.96 | 13.26 |
| | | NPK soil fertilization | 20 | 57.10 b | 6.05 | 10.60 |
| | | Foliar fertilization | 20 | 57.30 b | 6.48 | 11.31 |
| | | Mixed fertilization | 20 | 61.50 c | 4.98 | 8.10 |
| Leaf number | | Unfertilized | 20 | 21.10 a | 3.42 | 16.19 |
| | | NPK soil fertilization | 20 | 22.10 a | 2.59 | 11.74 |
| | Herbal extracts | Foliar fertilization | 20 | 21.80 a | 2.42 | 11.10 |
| | | Mixed fertilization | 20 | 23.00 ab | 3.13 | 13.60 |
| Fruit number | | Unfertilized | 20 | 5.80 a | 1.36 | 23.47 |
| | | NPK soil fertilization | 20 | 6.15 a | 1.14 | 18.48 |
| | | Foliar fertilization | 20 | 6.55 a | 1.15 | 17.50 |
| | | Mixed fertilization | 20 | 6.60 a | 1.27 | 19.29 |
| Chlorophyll, SPAD | | Unfertilized | 20 | 53.60 a | 4.92 | 9.19 |
| | | NPK soil fertilization | 20 | 54.95 a | 4.12 | 7.50 |
| | | Foliar fertilization | 20 | 54.50 a | 5.30 | 9.72 |
| | | Mixed fertilization | 20 | 56.55 b | 3.87 | 6.85 |

N—number of plants; X—average; s—standard deviation; CV—coefficient of variation; *t*-test ($p < 0.05$); the differences between any two averages are significant, if their values are followed by letters, or groups of different letters.

### 3.2.3. Treatment with Herbal Extracts

Similarly with the results corresponding to untreated and conventionally treated variants, the lowest averages of all analyzed traits correspond to the unfertilized variant. Differences are reported in the average foliar area of unfertilized plants on the one hand, and NPK-soil-fertilized, foliar-fertilized, and plants mixed-fertilized on the other hand. No statistical difference in average foliar areas is observed between NPK-soil-fertilized and foliar-fertilized plants and between unfertilized, NPK-soil-fertilized and foliar-fertilized plants for average leaf number and average chlorophyll content. Statistical differences are observed in the leaf number of plants unfertilized and mixed fertilized, and in the chlorophyll content of mixed-fertilized plants and the other three fertilization variants. No statistical differences are reported function of fertilization variant in fruit number (Table 2).

Overall, this study shows that the largest leaf areas were achieved with mixed soil and foliar fertilization in all three treatments, untreated (60.4 cm$^2$), conventionally treated (62.55 cm$^2$) and treated with herbal extracts (61.5 cm$^2$). Regardless of treatment, statistical differences are observed. In terms of leaf area, mixed soil and foliar fertilization represents the most suitable technological approaches in order to obtain large leaf areas. Lower results are reported for NPK soil fertilization and absence of fertilization (with a minimum average of 52.5 cm$^2$ corresponding to the application of the herbal extracts treatment). Results superior to these, but inferior to mixed fertilization are reported both when NPK soil fertilization and foliar fertilization were applied, regardless of the treatments administered. The highest averages, in this case, correspond to the experimental variants NPK soil and foliar fertilized, and conventionally treated, 58.1 cm$^2$ and 58.3 cm$^2$, respectively. The foliar area and leaf number have average variability, while chlorophyll content shows low to average variability, whatever fertilization type or treatment. The average fruit number has high variability for all experimental conditions, but because the values of the coefficient of variation are under 30% [43], the averages are significant (Table 2).

Sajid et al. [47] reported, under the conditions of an experiment where they applied foliar fertilization, a number of leaves/plant and fruits/plant located in the ranges of 40–60 leaves and 20–30 fruits, different from those obtained in a previous experiment (20.45–23.95 leaves and respectively 5.65–7.25 fruits). Saleem et al. [48] under the conditions of an experiment where they used soil fertilization, obtained a number of fruits/plant located in the range of 48 fruits—95 fruits, also different from the one obtained in the present experiment (5.60 fruits—6.55 fruits). It is believed that the differences regarding the number of leaves and flowers in the values presented in the present research and those reported by the aforementioned studies are due to the experimental conditions, namely the use of different tomato cultivars, in the pedo-climatic conditions specific to the Punjab regions and Faisalabad in Pakistan. For chlorophyll, the highest content (58.80 SPAD) is reported in conventionally treated and NPK mixed-fertilized plants, and the lowest content (53.60 SPAD) in unfertilized plants treated with herbal extracts. Higo et al. [49] reported a slightly lower chlorophyll content than that reported in the present study, in the range of 48.90–51.50 SPAD under soil fertilization with a fertilizer containing different doses of phosphorus, and Kazemi [50] obtained much lower values in the range of 13.12–25.14 SPAD.

The lowest averages of fruit weight, NUE, and dry matter differences are reported in untreated unfertilized plants (Table 3).

**Table 3.** The evolution of some production traits in the cv. Ruxandra variety, and function of agricultural inputs.

| Trait | Phytosanitary Treatment | Fertilization | N | X | s | CV (%) |
|---|---|---|---|---|---|---|
| Fruit weight, g | | Unfertilized | 20 | 188.45 a | 6.85 | 3.63 |
| | | NPK soil fertilization | 20 | 190.70 a | 6.08 | 3.19 |
| | | Foliar fertilization | 20 | 190.00 a | 5.87 | 3.09 |
| | | Mixed fertilization | 20 | 193.20 b | 5.91 | 3.06 |
| NUE, % | Untreated | Unfertilized | 20 | 0.18 a | 0.02 | 11.12 |
| | | NPK soil fertilization | 20 | 0.34 b | 0.08 | 23.51 |
| | | Foliar fertilization | 20 | 0.32 b | 0.07 | 22.80 |
| | | Mixed fertilization | 20 | 0.27 b | 0.02 | 8.57 |
| Dry matter, g | | Unfertilized | 20 | 5.70 a | 0.80 | 14.06 |
| | | NPK soil fertilization | 20 | 6.41 a | 0.45 | 7.03 |
| | | Foliar fertilization | 20 | 6.21 a | 0.59 | 9.43 |
| | | Mixed fertilization | 20 | 6.59 a | 0.67 | 10.12 |
| Fruit weight, g | | Unfertilized | 20 | 188.90 a | 5.69 | 3.01 |
| | | NPK soil fertilization | 20 | 196.10 b | 4.10 | 2.09 |
| | | Foliar fertilization | 20 | 193.60 b | 5.04 | 2.60 |
| | | Mixed fertilization | 20 | 193.50 b | 6.51 | 3.36 |
| NUE, % | Conventional | Unfertilized | 20 | 0.28 a | 0.03 | 10.71 |
| | | NPK soil fertilization | 20 | 0.35 b | 0.05 | 13.26 |
| | | Foliar fertilization | 20 | 0.33 b | 0.04 | 11.95 |
| | | Mixed fertilization | 20 | 0.29 a | 0.04 | 17.35 |
| Dry matter, g | | Unfertilized | 20 | 5.52 a | 0.63 | 11.42 |
| | | NPK soil fertilization | 20 | 6.41 a | 0.66 | 10.34 |
| | | Foliar fertilization | 20 | 6.14 a | 0.52 | 8.39 |
| | | Mixed fertilization | 20 | 6.65 b | 0.74 | 11.18 |
| Fruit weight, g | | Unfertilized | 20 | 188.50 a | 6.13 | 3.25 |
| | | NPK soil fertilization | 20 | 190.20 a | 5.05 | 2.66 |
| | | Foliar fertilization | 20 | 189.60 a | 5.25 | 2.77 |
| | | Mixed fertilization | 20 | 189.90 a | 5.03 | 2.65 |
| NUE, % | Herbal extract | Unfertilized | 20 | 0.18 a | 0.04 | 24.10 |
| | | NPK soil fertilization | 20 | 0.36 b | 0.03 | 9.66 |
| | | Foliar fertilization | 20 | 0.35 b | 0.04 | 10.92 |
| | | Mixed fertilization | 20 | 0.35 b | 0.04 | 10.16 |
| Dry matter, g | | Unfertilized | 20 | 5.58 a | 0.82 | 14.75 |
| | | NPK soil fertilization | 20 | 6.19 a | 0.68 | 11.06 |
| | | Foliar fertilization | 20 | 6.11 a | 0.57 | 9.41 |
| | | Mixed fertilization | 20 | 6.37 a | 0.70 | 10.94 |

N—number of plants; X—average; s—standard deviation; CV—coefficient of variation; *t*-test ($p < 0.05$); the differences between any two averages are significant, if their values are followed by letters, or groups of different letters.

### 3.2.4. Untreated Plants

There are no significant differences in the average fruit weights of unfertilized, NPK-soil-fertilized, and foliar-fertilized plants. The average fruit weight of mixed-fertilized plants differs from averages corresponding to unfertilized, NPK-soil-fertilized, and foliar-fertilized plants. Differences are reported in NUE averages of unfertilized plants, on the one hand, and that of NPK-soil-fertilized, foliar-fertilized, and mixed-fertilized plants, on the other hand. No statistical differences in average NUE are observed between NPK-soil-fertilized, foliar-fertilized, and mixed-fertilized plants. Concerning average dry matter, no statistical differences are observed function of fertilization variant.

### 3.2.5. Conventional Phytosanitary Treatment

Differences are reported in the average fruit weight of unfertilized plants on the one hand, and that of NPK-soil-fertilized, foliar-fertilized, and mixed-fertilized plants, on the other hand. No statistical differences in average fruit weight are observed between NPK-soil-fertilized, foliar-fertilized, and mixed-fertilized plants. NUE averages differ between NPK-soil-fertilized and foliar-fertilized plants, on the one hand, and unfertilized and mixed-fertilized plants, on other hand. There are no statistical differences in averages of NUE reported for unfertilized and mixed-fertilized plants, and also between NPK-soil-fertilized and foliar-fertilized plants. No differences are recorded in average dry matter of unfertilized, NPK-fertilized and foliar-fertilized plants, but significant differences are observed in average dry matter corresponding to the above-mentioned fertilization variants, and average dry matter of mixed-fertilized plants.

### 3.2.6. Treatment with Herbal Extracts

Regardless of the fertilization variant, no differences are observed in average fruit weight. For the same treatment, the same situation is also reported for plant dry matter. Similar to the situation reported for NUE averages of untreated plants, differences are reported in average NUE of unfertilized plants, on the one hand, and average NUE of NPK-soil-fertilized, foliar-fertilized, and mixed-fertilized plants, on the other hand. No statistical differences in average NUE are observed between NPK-soil-fertilized, foliar-fertilized, and mixed-fertilized plants. Fruit weights are highly homogeneous, and this is demonstrated by the low values of the variability coefficients, in all experimental conditions (fertilization and treatment). Dry matter shows average variability in all experimental conditions. Dry matter shows average variability in all experimental conditions. When conventional treatment is applied, whatever fertilization is administered, NUE shows average variability. NUE has high variability when no treatment and herbal extracts treatment is applied, whatever fertilization is applied, but averages are significant because the values of the coefficient of variation are under 30% (Table 3).

The development of the average weight of tomato fruits ranges between 188.45 g (unfertilized, untreated plants) and 196.10 g (NPK soil fertilized, plants treated with herbal extracts). Saalem et al. [48] following the application of soil fertilization, obtained plant weight averages in the range of 37–74 g, which is lower than that obtained in the present experiment (188.45–196.10 g). Also, lower values than those obtained in the present study are reported following the application of foliar fertilization, by Anwar et al. on 6 tomato cultivars with average yields in the range of 37–74 g [51] and Ashraf et al. on 12 cultivars with average yields in the range 30.96–45.20 g [52]. Average NUE shows values corresponding to the 0.18% range (unfertilized untreated and unfertilized, plants treated with herbal extracts) and 0.36 (mixed fertilized, treated with herbal extracts). The present study shows that the highest average NUE (0.36%) corresponds to NPK soil fertilization, and herbal extracts treatment, and this emphasizes that these experimental conditions lead to the best valorization of agricultural inputs analyzed in this study (Table 3).

In order to identify the influence of morpho-quantitative characteristics on tomato production, cv. Ruxandra, the multiple correlations between the mentioned factors were calculated (Table 4).

Multiple correlation analyses of production and the main morpho-productive characteristics of cv. Ruxandra show positive relationships and range between medium and strong. They range between R = 0.429 (foliar-fertilized, untreated plants) with a representativeness equal to 18.40% and R = 0.782 (foliar-fertilized, plants treated with herbal extracts) with a representativeness of 61.20% (Table 4). These results indicate that when foliar fertilization is applied, regardless of the treatments used, the predictability of the influence of morpho-quantitative traits on Ruxandra tomato production is lower.

**Table 4.** The multiple correlations between production and morpho-quantitative traits in the cv. Ruxandra tomato variety.

| Experimental Variant | Regression Line | R | R² |
|---|---|---|---|
| V1 | $Y = 17.233 + 0.324X1 + 0.134X2 + 0.039X3 + 0.116X4 - 0.041X5 + 0.334X6 + 0.498X7$ | 0.458 | 0.235 |
| V2 | $Y = 40.555 + 0.591X1 + 0.249X2 + 0.482X3 + 0.126X4 - 0.043X5 + 0.756X6 + 0.223X7$ | 0.554 | 0.308 |
| V3 | $Y = 43.295 + 0.178X1 + 1.283X2 + 0.058X3 + 0.921X4 - 0.046X5 + 1.041X6 + 0.578X7$ | 0.751 | 0.564 |
| V4 | $Y = 12.504 + 0.179X1 + 0.206X2 + 0.221X3 + 0.247X4 - 0.085X5 + 0.157X6 + 0.375X7$ | 0.453 | 0.206 |
| V5 | $Y = 120.003 + 0.614X1 + 0.245X2 + 0.137X3 + 0.424X4 - 0.053X5 + 0.078X6 + 0.326X7$ | 0.595 | 0.354 |
| V6 | $Y = 50.083 + 0.328X1 + 0.465X2 + 0.236X3 + 0.317X4 - 0.021X5 + 0.183X6 + 0.729X7$ | 0.661 | 0.437 |
| V7 | $Y = 6.591 + 0.062X1 + 0.355X2 + 0.058X3 + 0.427X4 - 0.065X5 + 0.144X6 + 0.437X7$ | 0.429 | 0.184 |
| V8 | $Y = 4.575 + 0.238X1 + 0.148X2 + 0.067X3 + 0.114X4 - 0.209X5 + 0.254X6 + 0.4358X7$ | 0.741 | 0.549 |
| V9 | $Y = 18.046 + 0.472X1 + 0.259X2 + 0.135X3 + 0.329X4 - 0.055X5 + 0.179X6 + 0.562X7$ | 0.782 | 0.612 |
| V10 | $Y = 69.467 + 0.120X1 + 0.355X2 + 0.109X3 + 0.562X4 - 0.049X5 + 0.381X6 + 0.529X7$ | 0.574 | 0.455 |
| V11 | $Y = 7.159 + 0.470X1 + 0.041X2 + 0.523X3 + 0.079X4 - 0.032X5 + 0.209X6 + 0.591X7$ | 0.698 | 0.488 |
| V12 | $Y = 35.251 + 0.280X1 + 0.176X2 + 0.026X3 + 0.533X4 - 0.057X5 + 0.159X6 + 0.798X7$ | 0.711 | 0.505 |

V1—unfertilized, untreated; V2—unfertilized, conventional treatment; V3—unfertilized, herbal extracts treatment; V4—NPK soil fertilized, untreated; V5—NPK soil fertilized, conventional treatment; V6—NPK soil fertilized, herbal extracts treatment; V7—foliar fertilized, untreated; V8—foliar fertilized, conventional treatment; V9—foliar fertilized, herbal extracts treatment; V10—mixed fertilized, untreated; V11—mixed fertilized, conventional treatment; V12—mixed fertilized, herbal extracts treatment; Y—production; X1—leaf area; X2—number of leaves; X3—number of fruits; X4—fruit weight; X5—NUE; X6—chlorophyll; X7—dry matter.

### 3.3. P. infestans Levels of Infection in cv. Ruxandra

For each phytosanitary treatment, the level of infection with *P. infestans* is significantly higher when plants are not fertilized (Table 5). Furthermore, no difference is observed between NPK soil fertilization and foliar fertilization in each treatment. The variants with mixed fertilization show the significantly lower infestation levels in each treatment, but in the conventional treatment the infestation levels do not differ from plants that were NPK soil and foliar fertilized. When the herbal treatment was used, the infestations levels significantly differ from NPK soil and foliar fertilization on the one hand, absence of fertilization, and mixed fertilization, on the other hand (Table 5).

**Table 5.** The *P. infestans infection* levels in cv. Ruxandra, %.

| Phytosanitary Treatment | Fertilization | N | X(%) | s | CV (%) |
|---|---|---|---|---|---|
| Untreated | Unfertilized | 20 | 44.55 a | 5.76 | 12.94 |
| | NPK soil fertilization | 20 | 35.10 b | 6.42 | 18.30 |
| | Foliar fertilization | 20 | 36.75 b | 5.94 | 16.16 |
| | Mixed fertilization | 20 | 32.75 bc | 5.16 | 15.75 |
| Conventional | Unfertilized | 20 | 22.65 a | 1.81 | 8.01 |
| | NPK soil fertilization | 20 | 14.60 b | 2.56 | 17.55 |
| | Foliar fertilization | 20 | 17.25 b | 5.07 | 29.37 |
| | Mixed fertilization | 20 | 12.30 bc | 1.87 | 15.18 |
| Herbal extracts | Unfertilized | 20 | 24.70 a | 4.55 | 18.44 |
| | NPK soil fertilization | 20 | 18.80 b | 2.21 | 11.78 |
| | Foliar fertilization | 20 | 20.90 b | 2.61 | 12.51 |
| | Mixed fertilization | 20 | 14.00 c | 2.53 | 18.10 |

N—number of plants; X—average level of infection with *P. infestans*; s—standard deviation; CV—coefficient of variation; *t*-test ($p < 0.05$); the differences between any two averages are significant, if their values are followed by letters, or groups of different letters.

In untreated plants, the average intensities of infection range within the limits of 44.55% corresponding to unfertilized untreated plants and 32.75% corresponding to mixed foliar- and NPK-soil-fertilized plants. When conventional treatment was applied, the average intensities of infection range within 22.65% (unfertilized, untreated plants) and 12.30%, corresponding to mixed-fertilized plants. When treatment with herbal extracts was administered, the average intensities of infection range within 24.70%, corresponding to the

unfertilized plants and 14%, corresponding to mixed-fertilized plants. Average variability of infection levels is observed for untreated and experimental variants treated with herbal extracts, regardless of the fertilization type (Table 5).

Within each fertilization strategy the average levels of infection with *P. infestans* on plants were compared corresponding to different fertilization variants, within each treatment strategy (lack of treatment between the treatments strategies). Considering all types of fertilization, the infection level corresponding to untreated plants differs from the infection levels reported in conventional and herbal extracts treatments. No significant difference is observed in average infection levels corresponding to conventional and herbal extracts treatments. For untreated plants, differences are observed only in infection levels corresponding to absence of fertilization on the one hand, and other fertilization variants used in this study. When conventional treatments and herbal treatments are applied, no significant differences are recorded in infection levels corresponding to NPK soil and foliar fertilization. Differences in the above-mentioned levels of infection, and those observed in the absence of fertilization, and mixed fertilization are observed (Table 6). These findings emphasize the capacity of herbal extracts treatment to diminish the *P. infestans* infection level with an efficacy similar to the conventional treatment.

**Table 6.** The influence of fertilization and treatments on *P. infestans infection* levels in cv. Ruxandra, %.

| Fertilization | Treatments | | |
| --- | --- | --- | --- |
| | Untreated | Conventional | Herbal Extracts |
| Control | 44.55 a | 22.65 a | 24.70 a |
| NPK soil fertilization | 35.10 b | 14.60 b | 18.80 b |
| Foliar fertilization | 36.75 b | 17.25 b | 20.90 b |
| Mixed fertilization | 32.75 b | 12.30 c | 14.00 c |
| Average level of infection with *P. infestans* (%) | 37.28 a | 16.70 b | 19.60 b |
| CV(%) | 13.71 | 26.66 | 22.75 |
| LSD$_{5\%}$ | 3.012 | 1.265 | 2.004 |
| F | 16.395 ** | 6.314 * | 7.113 * |

CV—coefficient of variation; LSD—least significant differences; F—Fisher coefficient; * $F < 0.05$; ** $F < 0.01$; the differences between any two averages are significant, if their values are followed by letters, or groups of different letters.

To test the feasibility of applying Principal Component Analysis (PCA) for environmental factors and agricultural inputs, the KMO and Bartlett tests were applied in all cases, for which values above 0.500 and $p < 0$ were obtained, 01, which demonstrated the feasibility of performing the factor analysis.

Three main factors were identified, namely the treatment (F1), the climatic regime (F2) and soil fertility (F3). Because in all cases eigenvalues greater than 1 correspond to the first two mentioned factors (F1 and F2), only these were considered in the present analysis [43]. In unfertilized plants, the treatment is responsible for 53.01% of the variance, and the climatic regime for 42.18% of variance (Figure 1a).

The treatments, in the absence of fertilization, are positively correlated with the majority of environmental factors (temperature—1, relative air humidity—2, and precipitations—4), with an infection level corresponding to conventional (6) and herbal extracts (7) treatments. It is noted that, also in the absence of fertilization, climatic factors are correlated with each other except relative air humidity (2) and with the levels of infection corresponding to both lack and administration of treatments (Figure 1a). When NPK soil fertilization is administered, the treatment is responsible for 58.31% of the variance, and the climatic regime for 36.23% of it. The treatments are positively correlated with the levels of infection corresponding to conventional (6) and herbal extracts (7) treatments, temperature (1), relative air humidity (3), and wind velocity. The levels of infection corresponding to

conventional (6) and herbal extracts treatments (7), temperature (1), wind velocity (3), and precipitations (3) are positively correlated with climatic regimen (Figure 1b).

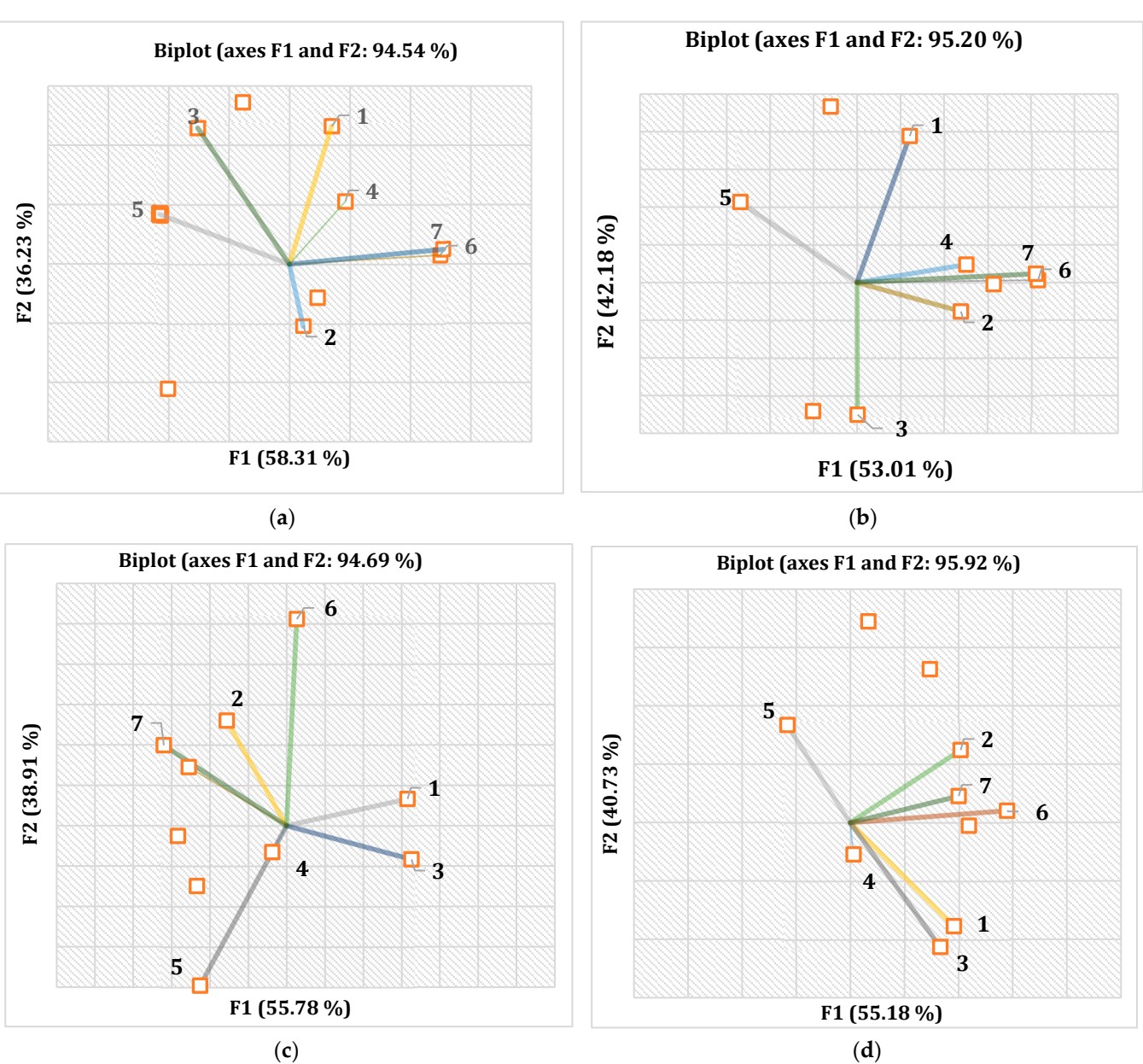

**Figure 1.** The representation in PC1 × PC2 plans of the variables corresponding to principal factors in context of the *Phytophthora infestans* (Mont.) de Bary attack and different agricultural inputs in cv. Ruxandra: (**a**) control unfertilized; (**b**) soil fertilization; (**c**) foliar fertilization; (**d**) mixed fertilization. 1—temperature (°C); 2—relative air humidity (%); 3—wind velocity (km/h); 4—precipitations (mm); 5—level of infection for untreated variant (%); 6—level of infection for conventionally treated variant (%); 7—level of infection for variant treated with herbal extracts.

Corresponding to foliar fertilization, the treatment is responsible for 55.78% of the variance, and the climate regime for 38.91% of variance. The treatments are correlated with the infection level corresponding to treatments (6 and 7), and also with temperature (1), and wind velocity (3). The climatic regimen is positively correlated with wind temperature (1), and relative air humidity (2), the infection level corresponding to conventional (6), and herbal extracts (7) treatments (Figure 1c). Similar interactions were identified by Litschmann et al. (2018) in a study performed in the Czech and Slovak Republics [53].

Moreover, when mixed soil fertilization is applied, the treatments are positively correlated with relative temperature (1), air humidity (2), wind velocity (3), level of infection corresponding to conventional (6) and herbal extracts (7) treatments. The climatic regimen is positively correlated with air relative humidity (2), both lack and administration of treatments (Figure 1d).

PCA shows that in absence of fertilization, treatments alongside the majority of climatic factors influence the infection level, when conventional and herbal extracts treatments are applied. When NPK soil fertilization and mixed fertilization are administered, treatments influence the infection level when conventional and herbal extracts treatments are applied, but climatic factors have a different influence on the level of infection.

Thus, it is mostly affected by all considered climatic factors (temperature, air relative humidity, wind velocity, and precipitations) when conventional treatment is applied, and by temperature, relative air humidity and wind velocity when herbal extracts treatment is administered. Foliar fertilization is characterized by the influence of conventional treatment on infection levels, together with temperature, and wind velocity. PCA also shows that alongside phytosanitary treatments, temperature, and relative humidity, there are climatic factors, which affect the levels of infection to the greatest extent.

## 4. Conclusions

According to the present study, production and morpho-productive characteristics of cv. Ruxandra are differently influenced by the fertilization strategy, while both treatment solutions, conventional and with herbal extracts, are efficient. In support of the mentioned assertion, it is highlighted that morphological traits (foliar area, leaf number, fruit number, and chlorophyll content), the production trait represented by dry matter, and levels of infection recorded the best results when mixed fertilization was performed. Fruit weight and NUE production traits recorded the best results when NPK soil fertilization was administered. Even though the averages of morpho-productive traits are superior when conventional treatment was applied, the differences compared to averages obtained when herbal extracts treatment is applied are not significant. Tomato production is strongly influenced by leaf area and number, NUE, chlorophyll content and fruit dry matter when herbal extracts treatment is administered in unfertilized, foliar- and mixed-fertilized plants, and when conventional treatment is administered together with foliar fertilization. The lowest average levels of infection correspond to conventional treatment, but they do not differ significantly from those reported when herbal extracts treatment is applied. The herbal extracts treatment has similar efficacy to diminish the *P. infestans* level of infection as conventional treatment, being mainly influenced by temperature, and relative air humidity, regardless of the fertilization strategies applied. The results of our study provide premises for the enhancement of tools for promoting sustainability.

**Author Contributions:** Conceptualization, R.A.S. and A.C.M.O.; methodology, C.-P.R., P.B. and C.M.M.; software, C.M.; validation, I.O. and A.C.M.O.; writing—draft preparation, R.A.S. and C.B.; supervision, A.C.M.O. All authors have read and agreed to the published version of the manuscript.

**Funding:** This research was partially funded by the project development of the skills of advanced and applied research in the STEAM+ Health, POCU/993/6/13/153310, project co-financed from the European Social Fund through the Human Capital Operational Program 2014–2020 and from PN-III-P4-ID-PCE-2020-1172 grant, number 243/2021.

**Institutional Review Board Statement:** Not applicable.

**Informed Consent Statement:** Not applicable.

**Data Availability Statement:** Data are contained within the article.

**Conflicts of Interest:** The authors declare no conflict of interest.

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
