# Peer review of "Morpho-Quantitative Traits and Interrelationships between Environmental Factors and Phytophthora infestans (Mont.) de Bary Attack in Tomato"

_sustainability, doi:10.3390/su16010301_

Round 1

Reviewer 1 Report

Comments and Suggestions for Authors

-Has Phytophthora been isolated or not?

-What is the environment in which it was developed?

-Is there a specific pathological scale that has been relied upon in estimating pathological degrees or not, and what is it?

-Was the infection caused by Phytophthora or was the infection natural?

-If the infection was artificial, how did it occur and at what age was the plant?

- Line No. 62: We write like fungus, not fungus.

- Line No. 98 refers to the year 2012.

- Line No. 401: Phytophthora infestans is written in italic.

Comments on the Quality of English Language

good

Author Response

Response to reviewer 1

Has Phytophthora been isolated or not?

Phytophtora was not isolated.

-What is the environment in which it was developed?

The natural environment as describel in article L.106-L111

-Is there a specific pathological scale that has been relied upon in estimating pathological degrees or not, and what is it?

No there is not.

-Was the infection caused by Phytophthora or was the infection natural?

The infection was natural.

-If the infection was artificial, how did it occur and at what age was the plant?

The infection was natural.

- Line No. 62: We write like fungus, not fungus.

Text was modified.

- Line No. 98 refers to the year 2012.

Text was modified.

- Line No. 401: Phytophthora infestans is written in italic.

Text was modified.

Reviewer 2 Report

Comments and Suggestions for Authors

Morpho-quantitative traits and interrelationships between environmental factors and Phytophthora infestans attack in tomato

The authors have performed a two-year field study where they compare three fertilization methods on morphology and quantitative traits of tomato, additionally they assessed these traits and the infestation of tomato leaves after infection with P. infestans while using either chemical control substance or herbal extracts in combination with fertilization variants.

General comments:

The paper is written in a very general way and in some parts (abstract; introduction; result) superficially, whereas in some parts basic information is missing. The authors should avoid repeating the facts in each chapter.

Abstract: is not sound; overall results are missing

Introduction: The authors don’t focus clearly on the main aspects that is the topic of their work.  

Materials and methods: The most important part for field experiment is the experimental design and that is missing. It is not clear with how many plants they worked and how the experiment was designed. It is not clear how they handled with the results of the two years. They don’t describe how they assessed the occurrence of P. infestans and plant infection.

The result part is written very monotonously and the findings are written in the discussion

All in all, since these main parts are missing I suggest to rewrite the manuscript and let it edit by a native English speaker.

Abstract: Please write in the abstract clear results of your experiments. The abstract is a kind of eye catcher if just write superficially the topic no one is interested in reading your paper

Keywords: I suggest Phytophthora infestans; herbal extracts instead of phytosanitary organic solution

Introduction:

The introduction part is very general and to some degree redundant. On the other hand, the aspects you deal with in your work, like climatic conditions and role of fertilization are briefly mentioned and no real insights on the influence of fertilization on morphological traits is mentioned in the introduction.

L 37: practices

L 43: do you mean largest?

L 43: occupy

L 37- 42: I suggest to reduce this part of first paragraph since this is not necessary for your topic

L 45: use

L 46-49: this information (except the information of FAO) is redundant, since they are not really relevant for your work.

L 55-58: I suggest to shorten the general sentences and specify to Phytophthora, your topic.

L 59-62: I doubt whether your reference is right for this statement; there are many publications that would fit better

L 62: pathogen

L 60-63: please write here some more detail on the climatic conditions that influence pathogen occurrence

L 69-72: please rewrite these sentence as ‘one may find’ ; ‘may have’ and ‘may be’ imply that there is no effect using essential oils.

L 71: fungi

L71-81: in this paragraph you line up the findings of some authors (Sarkhosh et al.; Seraluzzu et al.; Hussain et al.). Could you please explain more clearly the impact of the compounds on pathogenic fungi so one can follow why you used them for your work?

Please rewrite this paragraph and give deeper details of these references that refer to your work. Was the effect of compounds brought in connection with fungal diseases?  Even if you write antifungal effect then it would be interesting to know against which fungi. In the last paragraph you then can write that you used these compounds to reach your aim

L82-85: which fertilization and phytosanitary treatments?

L 84: environmental

L 85/92: I would rather use the word infection than attack (in the hole manuscript)

Materials and methods

In material and methods, the main information must be written and it doesn’t need to mention everything in two or three sentences beforehand, just describe shortly the key information.

Generally: you don’t need to write down the hole names of tomato and Phytophtora infestans, if you have mentioned it once. L. lycopersicum and P. infestans is enough.

L 89: what are the particularities of the location, please provide some key data (temperature, soil)

L 90: what are the experimental conditions?

L 92: please change the word ‘attack degree’; better level of infection or degree of infection

L 94: This sentence is redundant as you introduce the statistics separately

L97: it is not necessary to write about the seed producer etc.; just write cultivar Ruxandra (Agrosel, Romania): You can combine this with L 92, were you mention tomato crop, Solanum lycopersicum L. cv. Ruxandra (Agrosel, Romania), an early variety with medium vigour….

L99-101: it is enough to write cultivar Ruxandra or ‘Ruxandra’ (delete tomatoes belonging to the cultivar R.); please change it in whole text

L 101: please delete also

L 100: when did you fertilize and how often? In case of foliar fertilization at which stadium did you fertilize and how often?

L 101: experiment instead of experience; factor A

L102/105: please use treatment instead of grade

L105: how often did you treat the plants with conventional fungicides and herbal extracts?

L103: could you please write what the content of YaraVita mineral is and please write in brackets the company that produces it and in which country it is produced.

L 103-104: you give here the amount for foliar fertilization L/ha, but nowhere the experimental design is described: how many plants were investigated? How many plants were planted per area; how many replicates…How were the plants grown; at which stadium were plants fertilized, how was the watering,…

L 103: a4 = in which ratio was it mixed?

L105: factor B

L105: you can

L 107: which commercially available aqueous extracts? Herbal extracts?

L111: experiment

L123: Infection intensity and frequency of P. infestans were recorded …

L124: how did you calculate the infection degree and how the frequency? And on how many plants were disease assessed? And did you average the incidence in two years or calculate separately? All these basic information is missing in materials and methods.

L 130-131. This sentence is redundant

L130: please start a new paragraph for the statistic part

L149-151: these sentences belong to the paragraph before statistics

L 150-151: please name the available methodologies

Results:

L 154-157: These phrases are repeated (from material and methods) and can be deleted. Just start with the temperature from April to July in 2021 and in 2022

General question: in materials and methods you write that the experiments were performed in 2021 and 2022 respectively. Why do you give here average temperatures over both years together? The occurrence of Phytophthora is very climate dependent and averaging the temperature over many years would alter the results. 

L 158: The sentence: “For…obtained:” can be deleted (redundant), just start the sentence with: “The preciptitaion regime is…

L 170: You should try to describe the results paragraph briefly and concisely. Like: “Leaf surface of untreated and unfertilized plants show an average of 52.50 cm2.”

L 172: please add at the end of the first sentence the refence table (Table 2), so the reader knows where to look at.

L170-217: Actually, it is not necessary to read out the results in the table. It is more interesting to compare the results and actually you should compare for example the four fertilization treatments respectively for each trait and refer to the statistical difference (without underlining the 5 % threshold), for example: “the average foliar area was lowest in unfertilized and untreated plants; no difference in average foliar was seen between NPK soil fertilized and foliar fertilized untreated plants and untreated plants fertilized with mixed fertilizers showed the highest foliar area”. Thus, the reader is given a hint as to where to look exactly.

Table 2 and table 3: Please describe more precisely what is shown in the tables. Please replace the word ‘issue’ with ‘trait’; I would replace the word ‘unconventional’ by ‘herbal extracts’ (or s.th. like that; the meaning of unconventional is very broad): Please write below the tables the statistical analysis and the p-value.

Are the results shown here from the two years 2021-2022? How many plants did you assess per year? These are very important and basic information that miss in your materials and methods

L238: X2-number of leaves; X3 number of fruits

L239-241: suggestion: Multiple correlations analyses between production and main morpho-productive characteristics of cv. Ruxandra show positive relationships and range between medium and strong.

L248-262: Also, here in this paragraph it is more purposeful to compare the treatments and fertilzation varieties using the significant differences and not only confirm what is written in the table

L251: please write here the referring table number (Table 4).

In table 4 you show statistical analyses within each treatment. Did you also make a statistical analysis comparing the phytosanitary treatments (plus fertilization treatments) together? Since you have not shown any experimental design, it is not clear whether comparing treatments together is possible. From which year are these results?

Table 4: Please write here more precisely attack (infestation) degree is; is it in percent? Also, the abbreviations N (20 what? Leaves, plants, Experimental sections?), also s and CV should be explained. If you have performed two experiments in two years then it would be advisable to write the results of both years (2021 and 2022). It would be interesting to know how the occurrence and plant infection with P. infestans would fit to climatic conditions of the same year and not over two averaged years. If the results are very similar, then you have to argue it.

L271: eigenvalues

L272: please delete only (twice written)

Figure 1: L279-280: what is AD-control? What is GA-treatment? Foreign language; L 285: (b) should be (d); this figure is hardly to understand;

L284: ‘administered to soil and folia’ can be deleted

Discussion:

Many parts in your discussion are actually results and they would fit better in your result part; like lines 292-302; lines 337-342; lines 357-362; Lines 370-389. Of course, you discuss your results with other papers, but in this case try to only mention the result that fits to these comparisons. Or the other option is to combine Results and Discussion together.

L 337: rather ‘development’ than ‘evolution’

L 358: what is GA?

L396-398: This sentence is not complete; in conclusion you can drop the abbreviations F1 and F2

Comments on the Quality of English Language

edit by native English speaker is needed 

Author Response

Responses to Reviewer 2

Abstract: Please write in the abstract clear results of your experiments. The abstract is a kind of eye catcher if just write superficially the topic no one is interested in reading your paper.

The main results of the research are:

  • The use of mixed fertilization resulted in best performances of morphological (highest leaf area, highest number of leaves and fruits, highest chlorophyll content) and part of productive traits (highest content of dry matter), and lowest levels of infection in Ruxandra cultivar,
  • The use of NPK soil fertilization led to best performances of fruit weight and NUE, whatever administeed treatment.
  • The herbal treatment has similar efficacy to increase averages of morpho-productive traits, and diminish the infestans level of infection as conventional treatment.
  • The treatment efficacy is mainly influenced by temperature, and air relative humidity, whatever fertilization strategies applied.

and they are all included in Abstract.

Keywords: I suggest Phytophthora infestans; herbal extracts instead of phytosanitary organic solution

The suggestion is accepted, and replacement was made – L32.

 Introduction:

The introduction part is very general and to some degree redundant. On the other hand, the aspects you deal with in your work, like climatic conditions and role of fertilization are briefly mentioned and no real insights on the influence of fertilization on morphological traits is mentioned in the introduction.

L 37: practices

The correction was made – L38

L 43: do you mean largest?

The correction was made – L40

L 43: occupy

The correction was made – L41

L 37- 42: I suggest to reduce this part of first paragraph since this is not necessary for your topic

The paragraph was reduced.

L 45: use

The correction was made – L43

L 46-49: this information (except the information of FAO) is redundant, since they are not really relevant for your work.

The unnecessary text was eliminated.

L 55-58: I suggest to shorten the general sentences and specify to Phytophthora, your topic.

The paragraph is reformulated, and the topic is specific to P. infestans L49-L53.

L 59-62: I doubt whether your reference is right for this statement; there are many publications that would fit better

The citation is corrected – L53, L531.

L 62: pathogen

The correction was made – L41

L 60-63: please write here some more detail on the climatic conditions that influence pathogen occurrence

The paragraph is reformulated – L56-L62

L 69-72: please rewrite these sentence as ‘one may find’ ; ‘may have’ and ‘may be’ imply that there is no effect using essential oils.

The sentence is rewritten according to suggestions – L67-L69.

L 71: fungi

The paragraph was reformulated.

L71-81: in this paragraph you line up the findings of some authors (Sarkhosh et al.; Seraluzzu et al.; Hussain et al.). Could you please explain more clearly the impact of the compounds on pathogenic fungi so one can follow why you used them for your work?

The explanations are detailed in text – L73-L91.

Please rewrite this paragraph and give deeper details of these references that refer to your work. Was the effect of compounds brought in connection with fungal diseases?  Even if you write antifungal effect then it would be interesting to know against which fungi. In the last paragraph you then can write that you used these compounds to reach your aim.

The paragraph is rewritten – L73-L91, L92-L102

L82-85: which fertilization and phytosanitary treatments?

  • Foliar fertilization with a mineral complex, and soil fertilization with complex N15:P15:K15
  • Treatments with azoxistrobin, and lavender, thyme, and rosemary herbal extracts
  • The paragraph was rewritten – L92-L102

L 84: environmental

The correction was made – L97

L 85/92: I would rather use the word infection than attack (in the hole manuscript)

The suggestion was adopted.

Materials and methods

In material and methods, the main information must be written and it doesn’t need to mention everything in two or three sentences beforehand, just describe shortly the key information.

Generally: you don’t need to write down the hole names of tomato and Phytophtora infestans, if you have mentioned it once. L. lycopersicum and P. infestans is enough.

L 89: what are the particularities of the location, please provide some key data (temperature, soil)

The requested data are presented – L108-L111.

L 90: what are the experimental conditions?

The experimental conditions are mentioned L113-L114

L 92: please change the word ‘attack degree’; better level of infection or degree of infection

According to suggestion, the syntagm ‘level of infection’ is used in text.

L 94: This sentence is redundant as you introduce the statistics separately

The sentence was removed. Statistics is presented in a separate paragraph – L184-204.

L97: it is not necessary to write about the seed producer etc.; just write cultivar Ruxandra (Agrosel, Romania): You can combine this with L 92, were you mention tomato crop, Solanum lycopersicum L. cv. Ruxandra (Agrosel, Romania), an early variety with medium vigour….

The suggestion is adopted – L118.

L99-101: it is enough to write cultivar Ruxandra or ‘Ruxandra’ (delete tomatoes belonging to the cultivar R.); please change it in whole text

The suggestion is adopted.

L 101: please delete also

The word ‘also’ was deleted.

L 100: when did you fertilize and how often? In case of foliar fertilization at which stadium did you fertilize and how often?

Details are presented – L142-L148.

L 101: experiment instead of experience; factor A

The suggestion was adopted. L124

L102/105: please use treatment instead of grade

The suggestion was adopted. L126, L129.

L105: how often did you treat the plants with conventional fungicides and herbal extracts?

Details are presented – L142-L148.

L103: could you please write what the content of YaraVita mineral is and please write in brackets the company that produces it and in which country it is produced.

Details are presented – L146-L147.

L 103-104: you give here the amount for foliar fertilization L/ha, but nowhere the experimental design is described: how many plants were investigated? How many plants were planted per area; how many replicates…How were the plants grown; at which stadium were plants fertilized, how was the watering, …

The experimental design consists of 2 bifactorial experiments, as described L124-L141. The required details are presented L142-160.

L 103: a4 = in which ratio was it mixed?

The mixed fertilization consists of NPK soil fertilization, and foliar fertilization. They cannot be mixed in a ratio. Details are given in text – L127, L148-L151.

L105: factor B

Text was modified.

L105: you can

Text was modified.

L 107: which commercially available aqueous extracts? Herbal extracts?

Details are presented – L153-L156.

L111: experiment

The correction was made – L125

L123: Infection intensity and frequency of P. infestans were recorded …

Text was modified.

L124: how did you calculate the infection degree and how the frequency? And on how many plants were disease assessed? And did you average the incidence in two years or calculate separately? All these basic information is missing in materials and methods.

Details are presented – L169-L186.

The means of the climatic parameters, the morphometric traits, production traits of plants and the level of infection were calcualted by each experimental year. Because no significant differences (p>0.05%) between means were recorded, results are expressed as averages over the entire experimental period.

L 130-131. This sentence is redundant

The sentence was removed.

L130: please start a new paragraph for the statistic part

A new paragraph for statistics is started.

L149-151: these sentences belong to the paragraph before statistics

The sentence was introduced in suggested paragraph – L164.

L 150-151: please name the available methodologies

The methodologies are named. For pathogen attack – L169-L182, and for nitrogen use efficiency – L164-L166.

Results:

L 154-157: These phrases are repeated (from material and methods) and can be deleted. Just start with the temperature from April to July in 2021 and in 2022

The phrases were removed.

General question: in materials and methods you write that the experiments were performed in 2021 and 2022 respectively. Why do you give here average temperatures over both years together? The occurrence of Phytophthora is very climate dependent and averaging the temperature over many years would alter the results. 

The means of the climatic parameters, the morphometric traits, production traits of plants and the level of infection were calcualted by each experimental year. Because no significant differences (p>0.05%) between means were recorded, results are expressed as averages over the entire experimental period.

L 158: The sentence: “For…obtained:” can be deleted (redundant), just start the sentence with: “The preciptitaion regime is…

The sentence was removed.

L 170: You should try to describe the results paragraph briefly and concisely. Like: “Leaf surface of untreated and unfertilized plants show an average of 52.50 cm2.”

The text was modified.

L 172: please add at the end of the first sentence the refence table (Table 2), so the reader knows where to look at.

The text was modified.

L170-217: Actually, it is not necessary to read out the results in the table. It is more interesting to compare the results and actually you should compare for example the four fertilization treatments respectively for each trait and refer to the statistical difference (without underlining the 5 % threshold), for example: “the average foliar area was lowest in unfertilized and untreated plants; no difference in average foliar was seen between NPK soil fertilized and foliar fertilized untreated plants and untreated plants fertilized with mixed fertilizers showed the highest foliar area”. Thus, the reader is given a hint as to where to look exactly.

The text was modified according to suggestions.

Table 2 and table 3: Please describe more precisely what is shown in the tables. Please replace the word ‘issue’ with ‘trait’; I would replace the word ‘unconventional’ by ‘herbal extracts’ (or s.th. like that; the meaning of unconventional is very broad): Please write below the tables the statistical analysis and the p-value.

The text was modified according to suggestions.

The word issue was replaced with ‘Trait”, the term ‘unconventional’ was replaced with ‘herbal extracts’. The statistical analyze and p-values are mentioned below the tables.

Are the results shown here from the two years 2021-2022? How many plants did you assess per year? These are very important and basic information that miss in your materials and methods

The means of the the morphometric traits, production traits of plants were calcualted by each experimental year. Because no significant differences (p>0.05%) between means were recorded, results are expressed as averages over the entire experimental period.

The number of plants is 10 plants/variant/year (resulting in 20palnts/variant/by experimetal period presented in the article) is mentioned in L131-L133, L140-L141.

L238: X2-number of leaves; X3 number of fruits

Correction was made – L376.

L239-241: suggestion: Multiple correlations analyses between production and main morpho-productive characteristics of cv. Ruxandra show positive relationships and range between medium and strong.

Suggestion was adopted – L378-L380.

L248-262: Also, here in this paragraph it is more purposeful to compare the treatments and fertilzation varieties using the significant differences and not only confirm what is written in the table

The text was modified.

L251: please write here the referring table number (Table 4).

The number of table was mentioned.

In table 4 you show statistical analyses within each treatment. Did you also make a statistical analysis comparing the phytosanitary treatments (plus fertilization treatments) together? Since you have not shown any experimental design, it is not clear whether comparing treatments together is possible. From which year are these results?

According to suggestion, we made the proposed analyze, Table 6.

Table 4: Please write here more precisely attack (infestation) degree is; is it in percent? Also, the abbreviations N (20 what? Leaves, plants, Experimental sections?), also s and CV should be explained. If you have performed two experiments in two years then it would be advisable to write the results of both years (2021 and 2022). It would be interesting to know how the occurrence and plant infection with P. infestans would fit to climatic conditions of the same year and not over two averaged years. If the results are very similar, then you have to argue it.

The suggested corrections were made.

All tables had explanations of N (number of plants), CV – coefficient of variability.

L271: eigenvalues

The word was corrected – L441.

L272: please delete only (twice written)

The word was deleted – L442.

Figure 1: L279-280: what is AD-control? What is GA-treatment? Foreign language; L 285: (b) should be (d); this figure is hardly to understand;

The text was modified.

L284: ‘administered to soil and folia’ can be deleted

The words were deleted as suggested – L451.

Discussion:

Many parts in your discussion are actually results and they would fit better in your result part; like lines 292-302; lines 337-342; lines 357-362; Lines 370-389. Of course, you discuss your results with other papers, but in this case try to only mention the result that fits to these comparisons. Or the other option is to combine Results and Discussion together.

The text was modified.

We adopted the suggestion to combine the Results with Discussions – L211

L 337: rather ‘development’ than ‘evolution’

The replacement was made – L353.

L 358: what is GA?

The text was modified.

L396-398: This sentence is not complete; in conclusion you can drop the abbreviations F1 and F2

The text was modified.

Reviewer 3 Report

Comments and Suggestions for Authors

This is an interesting study and the authors have collected a unique dataset using cutting-edge methodology. The paper is generally well-written and structured and important for researchers who are related to this subject. Although of that, below I have provided some remarks:

The English language must be improved.

Material and Methods should be divided into more sections and supported by more references.

Line 26: of extracts correct to extract (delete of)

Line 47: Solanum should be in italic

Line 62: pathodens correct to pathogens

Line 71: fungus correct to fungi

Line 72: the reference should be written as a number

Line 77: The references should be written as numbers

Line 84: Environmental Factors correct to environmental factors

Line 85: Attack correct to attack

Line 96: lycopersycum correct to lycopersicum

Line 107: What are the components of the commercially available aqueous extracts 5%?

Lines 319-320: (20.45 – 23.95 leaves and 319 respectively 5.65 – 7.25 fruits) correct to (20.45 – 23.95 leaves and 5.65 – 7.25 fruits, respectively)

Lines 320-323: confused statement, please revise

Line 332: report correct to reported

Line 401: Phytophthora infestans should be in italic

Comments on the Quality of English Language

The English language must be improved.

Author Response

Response to reviewer 3

This is an interesting study and the authors have collected a unique dataset using cutting-edge methodology. The paper is generally well-written and structured and important for researchers who are related to this subject. Although of that, below I have provided some remarks:

The English language must be improved.

Material and Methods should be divided into more sections and supported by more references.

The Material and Methods weree divided into more sections and supported by more references, as suggested

Line 26: of extracts correct to extract (delete of)

Text was modified.

Line 47: Solanum should be in italic

Text was modified.

Line 62: pathodens correct to pathogens

Correction was made L53

Line 71: fungus correct to fungi

Text was modified.

Line 72: the reference should be written as a number

The references are written as numbers, but researchers names need to be mentioned.

Line 77: The references should be written as numbers

The references are written as numbers, but researchers names need to be mentioned.

Line 84: Environmental Factors correct to environmental factors

Correction was made L97

Line 85: Attack correct to attack

Text was modified.

Line 96: lycopersycum correct to lycopersicum

Correction was made L117

Line 107: What are the components of the commercially available aqueous extracts 5%?

Details are presented – L153-L156.

Lines 319-320: (20.45 – 23.95 leaves and 319 respectively 5.65 – 7.25 fruits) correct to (20.45 – 23.95 leaves and 5.65 – 7.25 fruits, respectively)

Text was modified.

Lines 320-323: confused statement, please revise

Text was modified.

Line 332: report correct to reported

Correction was made L303

 Line 401: Phytophthora infestans should be in italic

Text was modified.

Reviewer 4 Report

Comments and Suggestions for Authors

The work is important because it brings new elements in the tomato culture technology and I recommend writing the references in alphabetical order.

Author Response

Thank you. We have updated the references according the journal's layout.

Round 2

Reviewer 2 Report

Comments and Suggestions for Authors

MDPI Sustainability

Morpho-quantitative traits and interrelationships between environmental factors and Phytophthora infestans attack in tomato

The authors have thoroughly responded the critical points of the first review.

But there are still open questions concerning the statistical calculation and have to be cleared.

Furthermore, the English language should be improved and the manuscript can be further shortened by avoiding repetition, otherwise the reader will lose interest while reading.

Concerning the tables, it is important that all essential explanations are given correctly, so that the reader does not need to look up in the text all time.

The following changes should be considered

L 23: ‘(Mont.) de Bary’ can be deleted

L41: Solanum lycopersicum

L74: ‘results’ can be deleted

L75: relationship

L72-76: these sentences are written somehow contrary. Was the concentration rate of the essential oil highest when it completely inhibited mycelial growth? Then it should be written more like:  

Sarkhosh et al. ….. compounds. According to an in vitro trial they observed a linear relationship between Phytophthora palmivora mycelial growth and application rate of lavender essential oil. They found that administration of L. angustifolia essential oil at high concentrations completely inhibited P. palmivora mycelial growth [21].

Materials and Methods

L 147-152: For foliar fertilization Yara Vita mineral complex (Yara International; Norway) was administered in a dose of …

L 166: ..was calculated as function of fertilization…

L197: for calculating the significance of difference of what? In your work you have two factors. What did you test? And which test did you use for it? I’m still wondering why you do not compare the difference between the treatments concerning the traits. It would be more interesting whether there is a difference in treatment for foliar area, leaves number, fruits number and SPAD. As you made a bifactorial experiment this should be possible. And in this statistics paragraph you should mention on the basis of which test you performed ANOVA to find differences; for example did you use a t-test or Tukey test or which one?

Results

L233-265: Reading this part is very confusing. To have a better overview, you can split the results in three treatment-parts (suggestion see below) and then write the results according to the treatment so you don’t have to repeat all time the treatment in your text and you can really shorten your descriptions, like:

Untreated plants. The average foliar was lowest in unfertilized plants and highest in plants fertilized with mixed fertilizers (Table2). There was no difference in average foliar area when plants had NPK soil fertilization or foliar fertilization. ….

Conventional phytosanitary treatment. The lowest averages of all analyzed traits correspond to unfertilized variants. …

Treatment with herbal extracts. Similar to the above mentioned results the lowest averages of all analyzed traits correspond to unfertilized plants. ….

L 230-233: Please don’t use the word ‘is reported’ just write: there is no difference; or no difference is observed between…

L267-269: Overall, the study shows that the largest leaf areas were achieved with mixed fertilization, of soil and leaves, in all three treatments, untreated (60.4 cm2), conventional (62.55 cm2) and non-conventional (61.5 cm2)

L270-278: I cannot follow this conclusion that there is no statistical difference in the case of conventional and herbal extract treatments. In Table 2 also in untreated /mixed fertilization the leaf area has a significance of c and does not differ from the two other mentioned.

It is of course interesting to know whether leaf area was different when untreated, conventionally treated or treated with herbal extracts are compared. But how did you your statistical test for this? In Table 2 the explanation for this is missing and there must be at least different types of letters, if you made comparisons within the fertilizations of each treatment (factor A) and among treatments (factor B).

L289-291:

-        you don’t need to write here ANOVA (this self-evident), what is important is to mention the test you used when analyzing the data and comparing for example t-test or Tukey test plus the threshold for statistical difference (p < 0.05). I am wondering why you write two different p-values; it is not necessary since 0.01 is included in the 0.05 threshold.

-        ‘the difference between any two variants are significant..’ ; what do you mean by any two variants? Did you compare the means within each fertilization variety within each treatment? Or did you compare them statistically also among the treatments? Please write it  in the table captions more clearly.

L310..

I suggest also for these results to split the results in the three treatment parts to have a better overview (see above)

L353: please replace the word ANOVA by the test you made (see above).

L386-388: These results indicate that when foliar fertilization is applied, regardless of the treatments used, the predictability of the influence of morpho-quantitative traits on Ruxandra tomato production is lower.

Table 5: Please explain what X is; mean of what and which measuring? Why do aou differentiate the p values? And what is a-b<0.01%?

L389-420: Please sum up this very long paragraph. Just write down what is very special, like:

Within each phytosanitary treatment the level of infection with P. infestans is significantly highest when plants were not fertilized. Furthermore, no difference is seen between NPK soil fertilization and foliar fertilization in each treatment. The variants with mixed fertilization show the significantly lowest infestation levels within each treatment, but in the conventional treatment the infestation levels do not differ from plants that were NPK soil and foliar fertilized.

L416-4120: what do you mean by these sentences? I’m not sure whether these statements about average variability are necessary. This is sth. You also can see in the table.

L418: applied

Table 5: please write in table what X stands for; X = means is not enough for example:   X – average level of infection with P. infestans; in the table you should write X (%)

L425-435: Please delete applying LSD5% test as this is mentioned in Materials and Methods in the statistics part; but it is necessary to write it in the table caption. You can write:

Within each fertilization strategy the average levels of infection with P. infestans on plants were compared between the treatment strategies.

But you have not done it yet! You just compared the means over all fertilization variants, what is somehow not conclusive for me. It makes more sense to compare the single fertilization variants within the treatments (as you have done for the means totally).

Table 6: Please write in the table for what these data stand, like average level of infection with P. infestans (%)!

It is not necessary to write p>0.05%; the letter of significance tells already that there is no difference.

And please complete here your statistical calculations for each single fertilization variant.

L 504: results

L 506: applied (please correct this word in the whole text)

L512: significantly

Comments on the Quality of English Language

The English language should be improved 

Author Response

Morpho-quantitative traits and interrelationships between environmental factors and Phytophthora infestans attack in tomato

The authors have thoroughly responded the critical points of the first review.

But there are still open questions concerning the statistical calculation and have to be cleared.

The following changes should be considered

L 23: ‘(Mont.) de Bary’ can be deleted

The words were deleted as suggested, L24.

L41: Solanum lycopersicum

The correction was made in text, L41.

L74: ‘results’ can be deleted

The word was deleted as suggested.

L75: relationship

The correction was made in text.

L72-76: these sentences are written somehow contrary. Was the concentration rate of the essential oil highest when it completely inhibited mycelial growth? Then it should be written more like: 

Sarkhosh et al. ….. compounds. According to an in vitro trial they observed a linear relationship between Phytophthora palmivora mycelial growth and application rate of lavender essential oil. They found that administration of L. angustifolia essential oil at high concentrations completely inhibited P. palmivora mycelial growth [21].

The correction was made in text, as suggested, L72-76.

Materials and Methods

L 147-152: For foliar fertilization Yara Vita mineral complex (Yara International; Norway) was administered in a dose of …

The correction was made in text, as suggested, L147-L148.

The composition of Yara Vita mineral complex was specified because the reviewer asked for it, in first Revision, and now thinks is better to remove the text.

‘L103: could you please write what the content of YaraVita mineral is and please write in brackets the company that produces it and in which country it is produced.’

L 166: ..was calculated as function of fertilization…

The correction was made in text, as suggested. L165

L197: for calculating the significance of difference of what?

For calculating the significance of differences between the means of plants traits and levels of infections.

In text was added supplementary information.

In your work you have two factors. What did you test?

We tested the differences between the means of plants traits and levels of infections, function of fertilization (A), and phytosanitary treatments (B), L196-L197.

And which test did you use for it?

All tests are mentioned together with their use, L196-L206.

I’m still wondering why you do not compare the difference between the treatments concerning the traits. It would be more interesting whether there is a difference in treatment for foliar area, leaves number, fruits number and SPAD. As you made a bifactorial experiment this should be possible.

The differences between the treatments concerning the traits are compared L196-L197.

And in this statistics paragraph you should mention on the basis of which test you performed ANOVA to find differences; for example did you use a t-test or Tukey test or which one?

The information was added, L196.

Results

L233-265: Reading this part is very confusing. To have a better overview, you can split the results in three treatment-parts (suggestion see below) and then write the results according to the treatment so you don’t have to repeat all time the treatment in your text and you can really shorten your descriptions, like:

Untreated plants. The average foliar was lowest in unfertilized plants and highest in plants fertilized with mixed fertilizers (Table2). There was no difference in average foliar area when plants had NPK soil fertilization or foliar fertilization. ….

Conventional phytosanitary treatment. The lowest averages of all analyzed traits correspond to unfertilized variants. …

Treatment with herbal extracts. Similar to the above mentioned results the lowest averages of all analyzed traits correspond to unfertilized plants. ….

The text was modified as suggested, L229-L261.

L 230-233: Please don’t use the word ‘is reported’ just write: there is no difference; or no difference is observed between…

The text was modified as suggested.

L267-269: Overall, the study shows that the largest leaf areas were achieved with mixed fertilization, of soil and leaves, in all three treatments, untreated (60.4 cm2), conventional (62.55 cm2) and non-conventional (61.5 cm2)

The text was modified as suggested, L262-L264.

L270-278: I cannot follow this conclusion that there is no statistical difference in the case of conventional and herbal extract treatments. In Table 2 also in untreated /mixed fertilization the leaf area has a significance of c and does not differ from the two other mentioned.

The corrections are made in text, L264—L269.

It is of course interesting to know whether leaf area was different when untreated, conventionally treated or treated with herbal extracts are compared.

It is evident from table and comments (Table 2, L262-L264).

But how did you your statistical test for this?

As evident from the table, using t-test.

In Table 2 the explanation for this is missing and there must be at least different types of letters, if you made comparisons within the fertilizations of each treatment (factor A) and among treatments (factor B).

There are different letters (a, b, c).

L289-291:

-        you don’t need to write here ANOVA (this self-evident), what is important is to mention the test you used when analyzing the data and comparing for example t-test or Tukey test plus the threshold for statistical difference (p < 0.05). I am wondering why you write two different p-values; it is not necessary since 0.01 is included in the 0.05 threshold.

The reviewers’ suggestion are applied.

-        ‘the difference between any two variants are significant..’ ; what do you mean by any two variants? Did you compare the means within each fertilization variety within each treatment? Or did you compare them statistically also among the treatments? Please write it  in the table captions more clearly.

The corrections are made in captions as suggested.

L310..

I suggest also for these results to split the results in the three treatment parts to have a better overview (see above)

The text was modified as suggested.

L353: please replace the word ANOVA by the test you made (see above).

The replacement was made as suggested.

L386-388: These results indicate that when foliar fertilization is applied, regardless of the treatments used, the predictability of the influence of morpho-quantitative traits on Ruxandra tomato production is lower.

The text was modified as suggested, L373-L375

Table 5: Please explain what X is; mean of what and which measuring? Why do aou differentiate the p values? And what is a-b<0.01%?

We made the suggested corrections. We remove the differentiation of the p values.

 L389-420: Please sum up this very long paragraph. Just write down what is very special, like:

Within each phytosanitary treatment the level of infection with P. infestans is significantly highest when plants were not fertilized. Furthermore, no difference is seen between NPK soil fertilization and foliar fertilization in each treatment. The variants with mixed fertilization show the significantly lowest infestation levels within each treatment, but in the conventional treatment the infestation levels do not differ from plants that were NPK soil and foliar fertilized.

The replacement was made as suggested, L377-L382.

The paragraph was sumed up, L377-L384.

L416-4120: what do you mean by these sentences? I’m not sure whether these statements about average variability are necessary. This is sth. You also can see in the table.

The paragraph was removed.

We do not understand the meaning of the abbreviation ’sth’ used by reviewer. In context it doesn’t mean ‘something’.

It was added only at reviewer suggestion.

‘Table 4: Please write here more precisely attack (infestation) degree is; is it in percent? Also, the abbreviations N (20 what? Leaves, plants, Experimental sections?), also s and CV should be explained. If you have performed two experiments in two years then it would be advisable to write the results of both years (2021 and 2022). It would be interesting to know how the occurrence and plant infection with P. infestans would fit to climatic conditions of the same year and not over two averaged years. If the results are very similar, then you have to argue it.’

L418: applied

The paragraph was removed as suggested, so the word was not corrected.

Table 5: please write in table what X stands for; X = means is not enough for example:   X – average level of infection with P. infestans; in the table you should write X (%)

We added the suggested text, even though is more than evident from the title of the table.

L425-435: Please delete applying LSD5% test as this is mentioned in Materials and Methods in the statistics part; but it is necessary to write it in the table caption. You can write:

Within each fertilization strategy the average levels of infection with P. infestans on plants were compared between the treatment strategies.

The corrections were made according to suggestion, L398-L400.

But you have not done it yet! You just compared the means over all fertilization variants, what is somehow not conclusive for me. It makes more sense to compare the single fertilization variants within the treatments (as you have done for the means totally).

The means over all fertilization variants are compared (corresponding to first four lines – unfertilized, NPK, soil fertilization, foliar fertilization, mixed fertilization –). The single fertilization variants within the treatments are compared, L404-L410.

Table 6: Please write in the table for what these data stand, like average level of infection with P. infestans (%)!

We added the suggested text, even though is more than evident from the title of the table.

It is not necessary to write p>0.05%; the letter of significance tells already that there is no difference.

The text was removed.

And please complete here your statistical calculations for each single fertilization variant.

The statistical calculations for each single fertilization variant were added, Table 6.

L 504: results

The replacement was made as suggested, L481.

L 506: applied (please correct this word in the whole text)

The corrections were made.

L512: significantly

The correction was made, L490.